# Representation Learning on Native Cortical Surfaces: From Geometry to Individual Traits

## Abstract

Analyzing the intricate geometry of the cerebral cortex is fundamental to understanding the neuroanatomical basis of individual traits. However, the fundamental conflict between powerful, grid-dependent architectures like Transformers and the irregular cortical mesh has forced a compromise: the distortive practice of spherical projection. This act of simplification discards the geometric subtleties we aim to study. To resolve this foundational data-architecture mismatch, we propose the Native Cortical Surface Representation Learning Model (NCS-RL), an end-to-end framework that reshapes the data to fit the model, not the other way around. Its first component, the Canonical Surface Generator, creates a shared, regular topological grid across all subjects. Onto this grid, it precisely maps each individual's unique geometric details via diffeomorphic deformation. This single process achieves three critical goals simultaneously: it establishes a principled tokenization for Transformers, resolves inter-subject correspondence, and yields a spectrum of anatomically faithful variations for data augmentation. With the cortical surface now represented as a structured and geometrically rich sequence of tokens, the second component, the Cortical Transformer, is designed to interpret it. Its dual-pathway architecture is built to leverage this new data structure: one pathway uses our novel Adjacency Self-Attention to learn fine-grained local geometric patterns directly from the native surface priors, while the other captures global context. A gated mechanism then fuses these pathways, forging a holistic representation that understands not just what a cortical region is, but precisely how it is shaped. Moreover, to ensure geometric fidelity, our model was pre-trained on over $5,000$ subjects from the ABCD, HCP, and ABIDE datasets. Our method demonstrates state-of-the-art performance in experiments and ablation studies, including phenotype prediction and functional map regression. Our implementation is available in the supplementary material and will be released.

## 1 Introduction

The geometric form of the cerebral cortex, characterized by its intricate folding patterns, serves as a 'morphological fingerprint' linking neuro-anatomy to individual traits (e.g. behavioral characteristics, cognitive abilities, or functional MRI signal variations) Dale et al. (1999). However, deciphering this link is complex, hindered by the challenge of analyzing high-dimensional, noisy, and topologically irregular cortical surface meshes. This irregularity complicates the establishment of meaningful correspondence across subjects, and crucially poses a major challenge to the application of powerful deep learning architectures like Transformers Han et al. (2022), which thrive on structured, grid-like data. Developing a framework that can operate directly on cortical surfaces while resolving this irregularity is paramount for unlocking the information implicit in brain anatomy.

A major line of research prioritizes geometric fidelity by operating directly on native cortical meshes. While existing methods in this vein, such as Yang et al. (2024) convolutions, successfully extract local geometric features, their local receptive fields prevent them from capturing the holistic patterns of the cortex. However, deploying powerful global architectures, the natural solution, needs a universal topological blueprint that yields tokens comparable both within and across subjects. And the sheer diversity of cortical geometry makes tokenization on raw surface impossible Fawaz et al. (2021). For instance, the fundamentally incompatible structures of different anatomical regions within the same brain—like the elongated precentral gyrus and the complexly folded calcarine sulcus, cannot

Figure 1: An illustration of the deep-learning based cortical analysis method workflow.

be reconciled into a uniform patch topology Lohmann et al. (2008). This reveals the limitation of existing native-space methods: while anatomically precise, their nature as local feature aggregators precludes the extraction of higher-level semantic information. Overcoming this is our motivation.

To circumvent these tokenization and correspondence challenges, sphere-based methods became the dominant paradigm Zhao et al. (2023); Dahan et al. (2022b;a). By mapping cortices onto a common spherical domain, they establish a standardized topology that allows cortical data to be processed by architectures originally designed for images and grids, such as Vision Transformers. However, this solution is a critical compromise. The projection introduces non-linear geometric distortions, forcing models to learn from a surrogate, rather than the true, anatomy Bazinet et al. (2025). Crucially, this process decouples morphological features from their native geometric context. Critical measures like sulcal depth is preserved as a scalar value, but it loses its meaning as a descriptor of a real three-dimensional shape. Therefore, this work is dedicated to creating a Transformer that operates on the native cortical surface, free from the distortions of spherical projections.

To resolve this compromise between geometric fidelity and architectural power, we propose the **Native Cortical Surface Representation Learning Model** (NCS-RL). Our model operates directly on the native cortical surface, aiming to achieve high geometric fidelity and cross-subject comparability through a novel two-stage process, as illustrated in Fig. 1(b). The cornerstone of our model is the **Canonical Surface Generator**, a module designed to resolve the tokenization challenge. Adopting a template-first, personalization-second strategy, it first establishes a canonical topological foundation. Next, it precisely maps the fine-grained geometric details of each subject onto this regularized mesh. This process achieves two critical goals: it ensures high geometric fidelity while simultaneously yielding anatomically plausible samples for data augmentation. With the cortical surface now represented as a geometrically-rich sequence of tokens, the challenge is to interpret this data. Standard Transformer is geometry-agnostic. To overcome this, our **Cortical Transformer** employs a dual-pathway architecture. It runs our proposed Adjacency Self-Attention in parallel with standard self-attention to capture local geometric patterns and global semantic relationships, respectively. To ensure these pathways remain complementary, a mechanism for feature disentanglement and a subsequent gated fusion adaptively merge them to specialize the representation for the downstream task. Ultimately, this architecture learns not just what a cortical region is, but how it is intricately shaped.

In summary, our main contributions are as follows:

- We propose the Native Cortical Surface Representation Learning Model, an end-to-end framework that overcomes the core challenges of inter-subject comparability and geometric fidelity, paving the way for a new paradigm of large-scale, hypothesis-free discovery on cortical surfaces.

- We propose the Canonical Surface Generator, which decouples topological standardization from geometric personalization to simultaneously achieve robust inter-subject comparability and provide a principled source of anatomically plausible augmentation.

- We introduce the Cortical Transformer, whose expert-guided, dual-pathway architecture simultaneously learns geometric and semantic features to forge a link between neuroanatomy and cognitive or behavioral traits.

- Our method demonstrates state-of-the-art performance in experiments across phenotype prediction and functional activation map regression tasks.

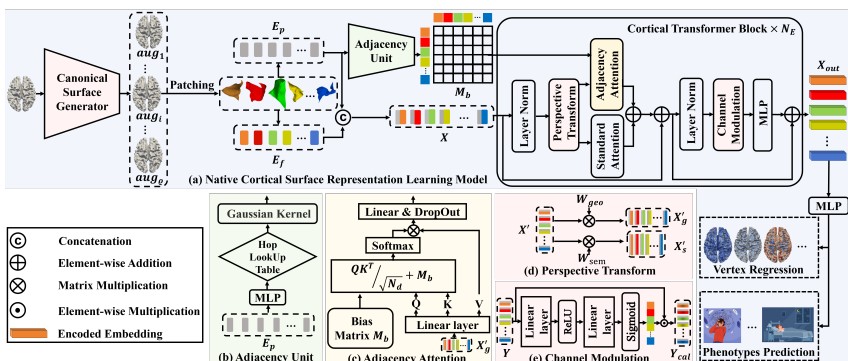

Figure 2: An overview of our framework.

## 2 METHOD

### 2.1 PRELIMINARIES AND OVERVIEW

Our NCS-RL employs a Masked Auto-Encoder (MAE) framework for brain mesh analysis (Fig. 2). Its Canonical Surface Generator (CSG) first normalizes each input mesh into a canonical form and creates augmentations. During pre-training, our Transformer learns to reconstruct the canonical mesh from masked patches, leveraging these augmentations for feature learning. For downstream tasks, we discard the decoder and use the pre-trained encoder to extract features from the canonical mesh for a final MLP predictor.

### 2.2 CANONICAL SURFACE GENERATOR

As illustrated in Fig. 3, our Canonical Surface Generator generates training samples through a three-stage pipeline that enforces a common topology while preserving subject-specific geometry.

#### 2.2.1 SIMPLIFICATION

We begin by simplifying a group-average surface to create a base mesh with a standardized topology. This base mesh acts as the uniform topological foundation for all subjects.

The process starts with a cohort of white matter surfaces, $C = \{I_i = (P_i, F)\}$, which are registered by Glasser et al. (2013), defined by the common face set $F$ and the vertex coordinate set $P_i$. From this cohort, we compute a group-average template, $I_T = \{P_T, F\}$, by averaging $P_i$. To preserve the points with significant anatomical features de Vareilles et al. (2023), we employ Fernández-Pena et al. (2023) to extract its cortical skeleton. This skeleton is represented by a vertex set, $P_{able}$, which corresponds to the sulcal fundi and gyral crowns. Based on these anatomical landmarks, we then construct a continuous field that quantifies the anatomical importance across the mesh surface.

To guide the simplification process, we introduce an adaptive cost function, $Cost(v)$, which balances geometric stability and anatomical importance of a vertex $v$:

$$Cost(v) = 0.5 \cdot C'_{geom}(v) + 0.5 \cdot C'_{anat}(v) \tag{1}$$

where $C'_{geom}$ and $C'_{anat}$ are the geometric and anatomical cost components, respectively. Each cost term is normalized with percentile ranking. The anatomical cost, $C_{anat}(v)$, is defined based on the geodesic distance of $v$ to the cortical skeleton $P_{able}$. We invert the ranking so that vertices closer to the skeleton receive a higher cost:

$$C_{anat}(v) = 1 - \text{PercentileRank}(\text{dist}_{geo}(v, P_{able})) \tag{2}$$

Here, $\text{dist}_{geo}(v, P_{able})$ is the shortest geodesic distance from $v$ to any vertex in the skeleton.

The geometric cost, $C_{geom}(v)$, ensures topological stability by penalizing distortion Lee et al. (1998). It is the ratio of the 1-ring neighborhood area after and before the removal of vertex $v$:

$$C_{geom}(v) = \frac{A_{new}(v)}{A_{old}(v)} \tag{3}$$

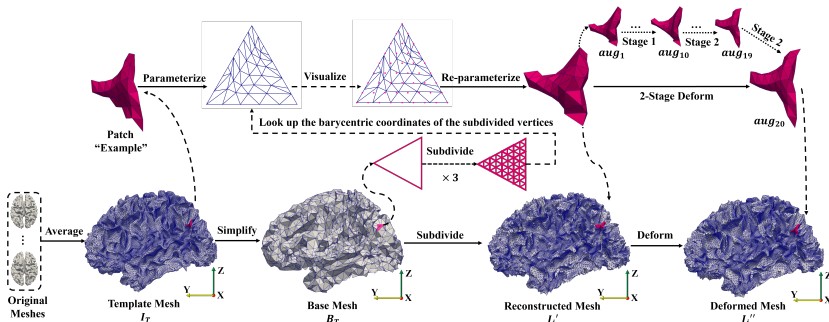

Figure 3: Illustration of Canonical Surface Generator.

To combine these two, before their summation, the raw $C_{anat}$ and $C_{geom}$ values are independently converted to their percentile ranks ($C'_{anat}$ and $C'_{geom}$), mapping them to a uniform $[0,1]$ interval.

With the cost function, we employ an iterative simplification algorithm described in Lee et al. (1998). Each iteration removes a vertex with the minimum cost. To re-triangulate the resulting hole, the vertex's neighborhood is projected onto a 2-D plane using a conformal map, and the new faces are generated with a constrained Delaunay triangulation. Concurrently, the global parameterization map $\Pi$ is updated by storing the barycentric coordinates of the removed vertex relative to these new faces, thus preserving a continuous mapping to the base mesh. The iterative algorithm is repeated until a target of $N_p = 1024$ faces is reached, which is chosen to balance geometric fidelity with the computational demands of model training. After that, we can get the base mesh $B_T$.

### 2.2.2 SUBDIVISION

Next, for each subject, we reconstruct a high-resolution surface $I'_i = (P'_i, F')$ that shares a common topology derived from the base mesh $B_T$ while preserving the subject's unique geometry.

To ensure the reconstructed topology is capable to represent the geometric details of $I_T$, we determine the subdivision level $d$ as the smallest integer satisfying $|P_{\text{sub}}(d)| \geq |P_T|$. Here, $|P_{\text{sub}}(d)|$ is the vertex count resulting from $d$ iterations of 1-to-4 Loop subdivision on the base mesh topology, and $|P_T|$ is the vertex count of the template $I_T$. With the subdivision level $d$ established, we generate a common high-resolution connectivity graph $F'$. This is achieved by applying $d$ iterations of a 1-to-4 Loop subdivision scheme to the connectivity of $B_T$. This operation is topological, refining the graph without computing new 3-D vertex positions. The result is a uniform face set $F'$ that is shared by all reconstructed surfaces, guaranteeing their topological equivalence.

For recovering each subject $i$'s detail, we individualize the base mesh, $B_i = (P_{B,i}, F_T)$, where the vertex set $P_{B,i}$ is populated by extracting the corresponding vertices from the subject's original surface $I_i$ via $I_{B,i} = \{\mathbf{v}_k \in P_i \mid k \in \mathcal{K}_{base}\}$, with $\mathcal{K}_{base}$ being the vertex indices from the original template preserved in $B_T$. This individualized base mesh $B_i$ establishes the direct correspondence between the 2-D parameter domain and the vertex indices of the subject's original mesh $I_i$. Following Lee et al. (1998), we determine the spatial coordinates of the interpolated vertices in $F'$. Specifically, for each interpolated vertex, we first compute its 2-D coordinate, $q_s$, as the midpoint of its parents' coordinates. We then locate its enclosing triangle within the 2-D parameterization established by $\Pi$, which allows us to identify the corresponding three vertices ,$\{v_{j,i}, v_{k,i}, v_{l,i}\} \subset V_i$, on the subject's original mesh. Next, we calculate the barycentric weights $(\alpha, \beta, \gamma)$ of $q_s$ with respect to this triangle's vertices. These weights are then used to interpolate the 3-D position:

$$v'_{s,i} = \alpha v_{j,i} + \beta v_{k,i} + \gamma v_{l,i} \tag{4}$$

The resulting collection of vertices forms the subject-specific vertex set $P'_i$. Finally, we assemble the reconstructed mesh $I'_i = (P'_i, F')$ by combining $P'_i$ with the common face set $F'$.

### 2.2.3 DEFORMATION

The subdivision, while establishing topological consistency, yields a surface $I'_i$ that filters the high-frequency geometric details of $I_i$. To address this, We model the deformation from $I'_i$ to $I_i$ as a

geodesic flow within Ceritoglu et al. (2013), implemented using Bône et al. (2018). This deformation is driven by optimizing a time-varying velocity field that minimizes the energy functional:

$$O[v] = \int_0^1 \|v(t)\|_L^2 \, dt + \frac{1}{\sigma_d^2} \text{Dist}(\phi_v(1, I_i'), I_i) \tag{5}$$

To capture complex cortical features, we employ the Varifold distance for the data attachment term, $\text{Dist}(\cdot, \cdot)$. Unlike point-based metrics, the Varifold distance is sensitive to both face position and orientation, which is critical for aligning the sharp crests of gyri and the deep fundi of sulci.

To ensure convergence, we implemented a two-stage optimization strategy. Specifically, the process begins with a small deformation kernel, $\sigma_{v,1} = L_{avg}(I_i')$, where $L_{avg}(I_i')$ is the average mesh edge length, and then transitions to a larger kernel ($\sigma_{v,2} = 10 \times L_{avg}(I_i')$) for local refinement. A constant data fidelity was used throughout both stages, with the noise standard deviation term in the energy functional set to $\sigma_d = 0.001$. The optimization is performed using L-BFGS and terminates on energy convergence ($\epsilon = 10^{-4}$) or after 200 iterations. The output for each subject is $I_i''$ that serves as a high-fidelity approximation of the original $I_i$. We sample $\varrho = 20$ time-points along the deformation path to generate a sequence of topologically-identical surfaces $\{aug_i\}$. This sequence, capturing a plausible anatomical evolution, constitutes our augmented dataset for pre-training.

### 2.3 Cortical Transformer

#### 2.3.1 Surface Embedding

Constrained by GPU memory and the computational complexity of the self-attention mechanism, we use the base faces of the template surface $B_T$ (from Section 2.2.1) to form our encoding units, which we refer to as patches. This design ensures that each patch consistently contains a fixed set of $N_v = 45$ vertices and $N_f = 64$ faces. Each patch is then encoded via two components: a feature embedding and a positional embedding. For the feature embedding, to enrich the representation and enhance its resolution, we define features at the face level. Specifically, for each face within the $j$-th patch, we compute a 10-dimensional feature vector, $z_{j,i}$ (face area, face normal vector, three internal angles, and three inner products between the face normal and the normals of its vertices). For the positional embedding, we use the centroid of the patch, $c_j$, to uniquely identify its spatial location. Finally, the feature and positional embeddings for the $j$-th patch are generated by multilayer perceptrons (MLP), formulated as $E_f = \text{MLP}(\{z_{j,i}\}_{i=1}^{N_f})$ and $E_p = \text{MLP}(c_j)$.

#### 2.3.2 Adjacency Unit

The Adjacency Unit is designed to generate a hop bias matrix, denoted as $M_b \in \mathbb{R}^{N_p \times N_p}$, from the spatial coordinates of visible patches. This process begins with a given batch of center centroids $\{c_j\}_{j=1}^{N_p}$ for visible patches, where a pre-trained and frozen Locator layer is first used to determine an ID for each patch. This can be understood as the locator mapping each patch to a predefined standard location. Following this, the pairwise hop counts between the input patches are retrieved from a hop look-up table, which stores the hop counts between any two standard locations. Finally, a parameterized Gaussian function, $a \cdot \exp(-x^2/(2\delta^2)) + b$, is used to map these hop counts (x) to their final values, which constitute the attention bias matrix $M_b$. The amplitude $a$, standard deviation $\delta$, and offset $b$ are learnable parameters. For cases with a hop count of $-1$, this value is replaced by a large number, causing the resulting bias to approach $b$ and thus applying a strong negative bias.

#### 2.3.3 Cortical Transformer Block

We introduce the Cortical Transformer Block to collaboratively process both global semantic correlations and local geometric structures. Its architecture (Fig. 2(a)) integrates standard and Adjacency Self-Attention, a Perspective Transform Layer, a Channel Modulation Layer, and an MLP.

Let the input embedding be $X \in \mathbb{R}^{N_p \times N_d}$, representing a set of $N_p$ tokens. Initially, the input $X$ undergoes a first Layer Normalization, yielding $X'$. This normalized feature is then fed into the Perspective Transform Layer, which decouples the representation into two specialized perspectives using two learnable weight matrices: $X_s' = X'W_{sem}^T$ and $X_g' = X'W_{geo}^T$ (Fig. 2(d)). Here, $X_s'$ is the semantic perspective, optimized for capturing high-level content, while $X_g'$ is the geometric

perspective, optimized for perceiving spatial structures. Subsequently, $X_s'$ and $X_g'$ are channeled into their respective customized attention pathways. The standard self-attention takes $X_s'$ as input to compute global, content-based dependencies among all tokens, producing $X_s''$.

The Adjacency Self-Attention receives $X_g'$ and incorporates the bias matrix $M_b$ to compute attention with a geometric prior. As shown in Fig. 2(c), $X_g'$ is linearly transformed into Query, Key, and Value matrices, which are then reshaped to form $h$ attention heads: $\{Q_i\}_{i=1}^h, \{K_i\}_{i=1}^h, \{V_i\}_{i=1}^h$. Incorporating $M_b$, the attention output for the i-th head is calculated as:

$$\text{Attention}_i = \text{softmax}\left(\frac{Q_i K_i^T}{\sqrt{d_k}} + B\right) V_i \qquad (6)$$

The outputs from all heads are concatenated and passed through a linear projection layer to yield the final output, $X_g''$. For the outputs of the two pathways, $X_s''$ and $X_g''$, we employ a learnable gate parameter, $w_{gate}$, to dynamically compute their fusion weights: $w_{geo} = \varsigma \cdot \text{Sigmoid}(w_{gate})$, $w_{std} = 1 - w_{geo}$ where $\varsigma$ is a hyperparameter controlling the maximum influence of the geometric pathway. The fused feature representation is then $X_{mix} = w_{std} X_s'' + w_{geo} X_g''$. Notably, this fusion is applied to patch tokens; the cls token's output is derived solely from $X_s''$ to preserve the purity of its global perspective. The fused attention result, $X_{mix}$, is added to the block's input $X$ via a first residual connection, followed by a second Layer Normalization, resulting in $Y = \text{LayerNorm}(X + X_{mix})$.

The tensor $Y$ is fed into Channel Modulation layer for feature recalibration. As illustrated in Fig. 2(e), each token passes through a two-layer MLP with a bottleneck structure and a Sigmoid function to generate the modulation weights $W_{\text{mod}}$. The calibrated output $Y_{\text{cal}}$ is computed via element-wise multiplication as: $Y_{\text{cal}} = Y \odot W_{\text{mod}}$. To dynamically transform each token, the refined tensor $Y_{\text{cal}}$ is passed through an MLP. Finally, the output of the MLP is added to the sub-layer's initial input, $Y$, via a residual connection to produce the block's final output, $X_{out} = Y + \text{MLP}(Y_{cal})$.

## 3 EXPERIMENTS

### 3.1 EXPERIMENTAL SETUP

#### 3.1.1 DATASETS AND METRICS.

**Adolescent Brain Cognitive Development (ABCD)** consists of MRI scans from over 11,500 children aged 9-10 years across the USA. From this dataset, we utilized 4,000 subjects for pre-training.

**Human Connectome Project (HCP)** contains brain MRI scans from 1,200 healthy young adults, preprocessed using the Glasser et al. (2013). On this dataset, we organized two experiments: phenotype prediction and functional activation map prediction. For **phenotype prediction**, we selected four phenotypes: VSPLOT_TC (VTC), PercStress_Unadj (PS), AngAggr_Unadj (AA), Strength_Unadj (STR) from 805 subjects, randomly split into training ($n = 643$) and testing ($n = 162$) sets; following He et al. (2024), predictions were evaluated using the Pearson Correlation Coefficient (PCC) and Root Mean Square Error (RMSE). For **functional activation map regression**, we targeted the Work Memory (WM) task, averaged the fMRI signal across time to create a 3D activation map, and projected it onto the surface; following Ellis & Aizenberg (2022), we evaluated spatial similarity using PCC and prediction error using Mean Absolute Error (MAE).

**Autism Brain Imaging Data Exchange (ABIDE)** is a dataset of 1,112 individuals, from which we selected subjects with available Autism Diagnostic Observation Schedule (ADOS) scores. Our preprocessing involved feature extraction and feature harmonization to mitigate batch effects. The dataset was partitioned into training and validation sets by acquisition site. Following Moradi et al. (2017), we evaluated model performance via Leave-One-Site-Out, using the PCC, MAE, and $Q^2$.

#### 3.1.2 IMPLEMENTATION DETAILS.

In both pre-training and downstream tasks, we employ the AdamW optimizer with a weight decay of 0.05 and a batch size of 20. We utilized a cosine learning rate schedule with a 10-epoch warmup period. The implementation is available in supplement, and we will release the codebase on GitHub. While hemispheres were analyzed independently for most tasks, their features were concatenated for ADOS prediction to capture the integrated neurological signatures of Autism. In our effort to

Table 1: Phenotype prediction results. Best and Second best results are highlighted and underlined. B, L, and R denote predict with the bilateral, left, and right cortical surfaces, respectively.

| METHOD | Side | VTC | PS | AA | STR |
|---|---|---|---|---|---|
| | | RMSE | | | |
| Dahan et al. (2022a) | B | 0.169±0.028 | 0.157±0.025 | 0.232±0.016 | 0.116±0.015 |
| Li et al. (2021) | B | 0.201±0.360 | 0.195±0.028 | 0.309±0.034 | 0.133±0.026 |
| Chen et al. (2022) | B | 0.204±0.290 | 0.197±0.034 | 0.303±0.025 | 0.129±0.021 |
| Ooi et al. (2022) | B | 0.187±0.031 | 0.208±0.039 | 0.311±0.027 | 0.125±0.025 |
| He et al. (2022) | B | 0.195±0.033 | 0.193±0.031 | 0.291±0.038 | 0.136±0.029 |
| He et al. (2024) | B | 0.183±0.025 | 0.171±0.029 | 0.228±0.021 | 0.117±0.017 |
| Yang et al. (2024) | R | 0.184±0.032 | 0.195±0.051 | 0.255±0.048 | 0.115±0.029 |
| | L | 0.197±0.040 | 0.183±0.038 | 0.289±0.044 | 0.106±0.037 |
| Liang et al. (2022) | R | 0.176±0.051 | 0.165±0.029 | 0.259±0.055 | 0.110±0.018 |
| | L | $\underline{0.163 \pm 0.047}$ | 0.159±0.030 | 0.240±0.027 | 0.113±0.021 |
| ours | R | 0.166±0.023 | **0.147±0.036** | $\underline{0.225 \pm 0.019}$ | **0.094±0.021** |
| | L | **0.159±0.014** | $\underline{0.150 \pm 0.031}$ | **0.218±0.018** | $\underline{0.101 \pm 0.017}$ |
| | | PCC | | | |
| Dahan et al. (2022a) | B | 0.091±0.206 | 0.067±0.128 | 0.101±0.122 | $\underline{0.130 \pm 0.162}$ |
| Li et al. (2021) | B | 0.102±0.241 | 0.056±0.307 | 0.071±0.255 | 0.113±0.227 |
| Chen et al. (2022) | B | 0.098±0.315 | 0.061±0.324 | 0.088±0.262 | 0.131±0.274 |
| Ooi et al. (2022) | B | 0.111±0.372 | 0.053±0.333 | 0.086±0.349 | 0.122±0.305 |
| He et al. (2022) | B | 0.096±0.262 | 0.068±0.250 | 0.092±0.242 | 0.109±0.259 |
| He et al. (2024) | B | 0.122±0.227 | 0.073±0.291 | 0.105±0.273 | 0.133±0.264 |
| Yang et al. (2024) | R | 0.009±0.329 | 0.071±0.325 | 0.103±0.325 | 0.070±0.328 |
| | L | 0.011±0.320 | $\underline{0.081 \pm 0.320}$ | 0.054±0.328 | 0.082±0.327 |
| Liang et al. (2022) | R | **0.172±0.320** | 0.068±0.327 | -0.079±0.326 | 0.098±0.182 |
| | L | 0.118±0.324 | -0.238±0.312 | **0.108±0.325** | 0.088±0.328 |
| ours | R | $\underline{0.125 \pm 0.122}$ | 0.076±0.171 | $\underline{0.105 \pm 0.126}$ | 0.119±0.209 |
| | L | 0.113±0.209 | **0.081±0.207** | 0.101±0.152 | **0.135±0.197** |

establish a fair comparison, we faced a challenge: a scarcity of methods from the last three years that are open-source, relevant to our task, and verifiable with available data. Therefore, we feature only three comparable methods in our experiments on surface fMRI regression and ADOS prediction.

## 3.2 PHENOTYPE PREDICTION

For phenotype prediction, our method was benchmarked against state-of-the-art approaches (Table 1 Upper for RMSE, Lower for PCC). In terms of RMSE, our model achieves top-tier performance, ranking first or second in 7 of 8 hemisphere-based predictions. For instance, on the Strength (STR) task, our model achieves an RMSE of 0.094 (R), outperforming the Liang et al. (2022) baseline (0.110). Regarding PCC, our model's primary advantage is its robustness. Unlike Liang et al. (2022), which exhibits high volatility and learns erroneous negative correlations on certain tasks (e.g., PS, AA), our model consistently maintains positive correlations across all phenotypes. This avoidance of directionally incorrect predictions underscores its stability and reliability. We attribute this enhanced performance to two core architectural advantages. First, by outperforming Dahan et al. (2022a), our model confirms the importance of operating on the native cortical mesh, which avoids the geometric distortions of spherical projections. Second, its superiority over Liang et al. (2022) demonstrates that our bi-path attention design, by integrating multi-scale geometric information, is crucial for preventing overfitting to spurious correlations and thus enhancing prediction stability.

## 3.3 FMRI ACTIVATION REGRESSION

As shown in Table 2, our method achieves the highest spatial similarity, with a PCC of 0.562 (R). Furthermore, our method yields the lowest prediction error, with an MAE of 0.075 (R). To visualize the result, we randomly select a subject and display their ground truth and predicted fMRI signal

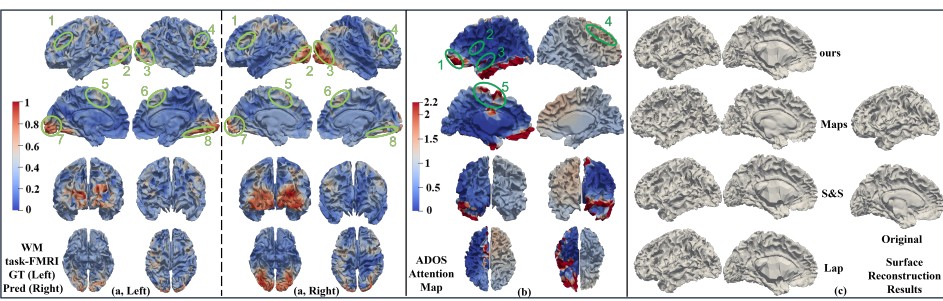

Figure 4: Visualizations of (a) FMRI activation map regression, (b) ADOS prediction model's attention map, and (c) the surface reconstruction results.

Table 2: Results of FMRI activation map regression. Best results are highlighted. B, L, and R denote predictions made using the bilateral, left, and right cortical surfaces, respectively.

| Method | Side | PCC | MAE |
|---|---|---|---|
| Ellis & Aizenberg (2022) | B | 0.550 | - |
| Yang et al. (2024) | R | $0.533 \pm 0.009$ | $0.078 \pm 0.04$ |
| | L | $0.527 \pm 0.007$ | $0.079 \pm 0.03$ |
| Liang et al. (2022) | R | $0.523 \pm 0.012$ | $0.079 \pm 0.06$ |
| | L | $0.514 \pm 0.013$ | $0.081 \pm 0.04$ |
| ours | R | $\mathbf{0.562 \pm 0.007}$ | $\mathbf{0.075 \pm 0.003}$ |
| | L | $0.537 \pm 0.005$ | $0.078 \pm 0.001$ |

intensities in Fig. 4(a) (left and right, respectively). We also point eight regions with notably similar signal intensity distributions using light green circles: 1-the left middle frontal gyrus, 2-the left superior and lateral occipital gyri, 3-the right superior and lateral occipital gyri, 4-the right middle frontal gyrus, 5-the left paracentral lobule, 6-the right paracentral lobule, 7-the left lingual gyrus, and 8-the right lateral occipitotemporal gyrus. And these regions are components of the networks responsible for working memoryYee et al. (2010).

## 3.4 ADOS PREDICTION

As shown in Table. 3, our proposed method achieves a superior performance on ADOS prediction task, yielding PCC scores of 0.59 on the PITT site and 0.42 on the NYU site. A key comparison is with the linear model of Moradi et al. (2017), which reveals a classic bias-variance trade-off. While their method achieves a lower MAE by using smoothed, low-dimensional features, our end-to-end model attains a significantly higher PCC by leveraging rich, high-dimensional geometric features directly from the cortical surface to capture more complex biological patterns. The positive $Q^2$ scores (0.18 and 0.26) decisively show the superiority of our low-bias strategy, indicating that the model has genuine predictive power, unlike other baselines which fail to generalize ($Q^2 <$ 0). This confirms that the significant gains in capturing the correct biological trend outweigh the moderately higher prediction variance. To investigate the neurobiological basis of these predictions, we visualized the attention map from the final layer of the fine-tuned model (shown in Fig. 4(b)). We also point out five important regions (colored in red, 1-left orbital gyrus, 2-left inferior frontal gyrus, 3-left superior temporal gyrus, 4-Right superior frontal gyrus, 5-Central anterior gyrus) with dark green circle, which are consistent with well-established Autism Spectrum Disorder-related biomarkersJacobson et al. (1988); Bauman (1991); Barnea-Goraly et al. (2004).

## 3.5 VALIDATE THE ANATOMICAL VALIDITY OF CSG.

Results in Table 4 highlight the trade-offs between accuracy and detail fidelity among methods. Our simplification&subdivision pipeline, $S\&S$, significantly improves upon the baseline Lee et al. (1998), reducing Average Surface Distance (ASD) from 1.520 to 0.773 and increasing Curvature Correlation (CC) over fourfold from 0.159 to 0.652. Other methods show limitations. While Sorkine et al. (2004) achieves the lowest ASD (0.766), its high Hausdorff Distance (HD) (10.128) indicates

Table 3: Results of ADOS prediction. Best results are highlighted.

| Method | Site | PCC | MAE | $Q^2$ |
|---|---|---|---|---|
| Moradi et al. (2017) | NYU | 0.22 | **1.57** | -0.04 |
| | PITT | 0.56 | **1.08** | 0.22 |
| Yang et al. (2024) | NYU | 0.06 | 3.62 | -0.38 |
| | PITT | 0.12 | 3.25 | -0.25 |
| Liang et al. (2022) | NYU | 0.18 | 3.45 | -0.15 |
| | PITT | 0.29 | 2.89 | -0.05 |
| **ours** | NYU | **0.42** | 2.81 | **0.18** |
| | PITT | **0.59** | 2.05 | **0.26** |

large local errors, a flaw our method mitigates (HD: 8.190). Conversely, while Lee et al. (1998) attains the best HD (7.791), it fails to capture geometric detail (CC: 0.159). In contrast, our method achieves the highest CC (0.676) while maintaining competitive HD and ASD values. This demonstrates an balance between local detail and overall accuracy, as visualized in Fig. 4(c).

Table 4: Results of surface reconstruction. The best results are highlighted.

| Method | HD | ASD | CC |
|---|---|---|---|
| Lee et al. (1998) | **7.791 ± 0.052** | 1.520 ± 0.011 | 0.159 ± 0.003 |
| $S\&S$ | 10.207 ± 0.061 | 0.773 ± 0.005 | 0.652 ± 0.002 |
| Sorkine et al. (2004) | 10.128 ± 0.068 | **0.766 ± 0.004** | 0.658 ± 0.003 |
| $ours$ | 8.190 ± 0.049 | 0.813 ± 0.005 | **0.676 ± 0.001** |

### 3.6 Deconstruction of the Cortical Transformer.

To validate our design, we conducted an ablation study for the left hemisphere WM-task prediction, with results presented in the right sub-table of Table 5. The results show a incremental benefit for each added component. Starting from the Liang et al. (2022) baseline (PCC: 0.514), adding the BA module boosts performance to 0.526, which is further improved to 0.532 with the PT module. Our model integrates these components to achieve the highest PCC of 0.537 and the lowest variance (±0.005), confirming that each part of our architecture is essential to its superior performance.

## 4 Conclusion

We propose the NCS-RL, a framework for cortical analysis. Its Canonical Surface Generator establishes cross-subject correspondence and augment surface, while its Cortical Transformer uses a dual-pathway attention to fuse local geometric details with global context. Experiments validate our approach, demonstrating superior performance across multiple tasks. However, a limitation emerged in our ADOS prediction experiment: the right hemisphere's attention map lacked the clear hotspots seen on the left (Fig. 4(b)). We attribute this to the lack of an inter-hemispheric communication mechanism in our model. Future work will therefore prioritize developing a cross-hemispheric attention to capture the complex signatures of neuro-developmental disorders.

Table 5: Results of deconstruction of Cortical Transformer. Best results are highlighted. MM is baseline Liang et al. (2022). BA is the Bi-Path Attenation. PT is the Persperctive Transform Layer. CM is the Channel Modulation Layer.

| Method | PCC |
|---|---|
| MM | 0.514 ± 0.013 |
| MM+BA | 0.526 ± 0.008 |
| MM+BA+PT | 0.532 ± 0.005 |
| MM+BA+CM | 0.527 ± 0.007 |
| ours | **0.537 ± 0.005** |

## 5 THE USE OF LARGE LANGUAGE MODELS STATEMENT

We use it on 2 ways: correct the code, polish the writing.

## 6 ETHICS STATEMENT

This work utilizes publicly available, de-identified data from the ABCD, HCP, and ABIDE datasets. Data collection for these original studies was conducted with appropriate institutional review board (IRB) approval and informed consent from all participants. Our research is foundational and intended for methodological advancement, not for clinical diagnosis.

## 7 REPRODICIBLITY STATEMENT

Our code is provided in the supplementary material and will be made publicly available on GitHub upon publication. The data used are publicly available from the ABCD, HCP, and ABIDE portals. All hyperparameters and training details required to reproduce our main results are detailed in Method section and the Experimental Setup subsection.

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
