# OpenReview forum: "REPRESENTATION LEARNING ON NATIVE CORTICAL SURFACES: FROM GEOMETRY TO INDIVIDUAL TRAITS"
_ICLR.cc/2026/Conference — Submitted to ICLR 2026_

### Official Review · Reviewer_PQk9 · 2025-10-29

**Soundness:** 3
**Presentation:** 3
**Contribution:** 3
**Rating:** 4
**Confidence:** 2

**Summary:**

This paper presents a new method for representation learning dedicated to cortical surface. The method is based on transformers and proposed a new way to use them with such surface.

**Strengths:**

The proposed framework is able to deal with the specificity of cortical surface. Especially the authors propose a technique to avoid deformation that appears with previous methods. Thus, the results are among the best compare to state-of-art methods. The main contribution is the full pipeline with dedicated modules. The framework is clearly described with all the important information. Since the cortical surface is important, and a central tool, in neuroscience the impact of a better reconstruction may be particularly noticeable in this community.

**Weaknesses:**

I see several weaknesses in such paper.

- The contribution is too focused on the cortical surface, and it is unclear if it may be extended to other kind of surface. Thus, only the neuroscience community may be interested in such paper, missing a large part of the ICLR community.
- There exists some well known bias or fairness issues with cortical data. For example, the site of acquisition is important, and the gender has some influence. What are their impact on the model?
- Please check how the references are inserted into the LaTeX file and use \citep for paper citation and \citet for title...

**Questions:**

My main question is about the bias management in such framework. How to deal with well known bias (see [1])?

- [1] Linhardt, D., Woletz, M., Paz‐Alonso, P. M., Windischberger, C., & Lerma‐Usabiaga, G. (2025). Biases in volumetric versus surface analyses in population receptive field mapping. Human Brain Mapping, 46(2), e70140.

---

### Official Review · Reviewer_TC8q · 2025-10-31

**Soundness:** 2
**Presentation:** 3
**Contribution:** 2
**Rating:** 4
**Confidence:** 4

**Summary:**

This paper introduces Native Cortical Surface Representation Learning (NCS-RL), which is designed to analyze the cerebral cortex's geometry directly on its native surface, avoiding distortions introduced by spherical projections. The framework consists of Canonical Surface Generator (CSG) and Cortical Transformer. CSG establishes a shared topological grid across subjects and maps individual cortical geometries onto it using diffeomorphic deformation, enabling consistent tokenization for Transformers and anatomically plausible data augmentation. Cortical Transformer adopts a dual-pathway Transformer architecture that uses adjacency self-attention for local geometric features, standard self-attention for global semantic context, and a gated fusion mechanism to integrate both pathways. The model is pre-trained on over 5,000 subjects from ABCD, HCP, and ABIDE datasets and evaluated on tasks including phenotype prediction, fMRI activation map regression, and Autism Diagnostic Observation Schedule (ADOS) prediction.

**Strengths:**

1. The CSG elegantly solves the challenge of tokenizing irregular cortical meshes by decoupling topology and geometry.

2. Unlike spherical projection methods, it preserves native cortical geometry, which is crucial for accurate neuroanatomical analysis.

3. The model is tested across multiple datasets and tasks.

4. The authors commit to releasing code.

**Weaknesses:**

The motivation of this work is to avoid distortions from spherical projection. However, does using a group-average template introduce similar drawbacks?

1. The model processes hemispheres independently, which may limit its ability to capture cross-hemispheric interactions.

2. The manuscript does not report the time consumption of the multi-stage pipeline (simplification, subdivision, deformation) or the Transformer architecture. The approach appears computationally intensive.

3. Comparisons are limited to a few baselines, which may not fully contextualize the model’s performance relative to other methods.

4. Some figures (e.g., Fig. 3) lack detailed captions or explanations, which could hinder understanding for readers less familiar with the domain.

5. Although the model demonstrates robustness, the high-dimensional features and augmentation strategy could pose a risk of overfitting without careful regularization.

6. The performance of the model using supervised learning without pretraining is not reported, making it impossible to assess the impact of pretraining.

**Questions:**

1. The model processes hemispheres independently, which limits its ability to capture cross-hemispheric interactions. You mentioned that this limitation could cause issues and proposed incorporating cross-hemispheric attention mechanisms in future work. Beyond the increased memory requirements, are there any other challenges or potential difficulties that might arise with this approach?

2. What about the training time, inference speed, and computational cost per subject?

3. While the work aims to avoid distortions from spherical projection, does using a group-average template introduce similar geometric drawbacks? Have the authors considered alternatives that could mitigate potential template-induced distortions?

4. The study compares to a limited set of baselines. Could the authors provide comparisons to additional state-of-the-art methods to better contextualize performance?

5. What are the performances of the model using supervised learning without pretraining?

6. Some figures (e.g., Fig. 3) are dense and lack detailed captions. Could the authors provide expanded explanations or visual guides to improve interpretability?

7. Did the authors explore regularization strategies (e.g., dropout, weight decay) to ensure robustness?

---

> ### Author Response · Authors · 2025-11-13
>
> thanks for your comments. All your proposed problems are valid, and it would help us to improve our manuscript. big thanks!

---

### Official Review · Reviewer_7orw · 2025-10-31

**Soundness:** 1
**Presentation:** 2
**Contribution:** 2
**Rating:** 2
**Confidence:** 5

**Summary:**

This paper introduces the Native Cortical Surface Representation Learning Model (NCS-RL), a method designed to analyse cortical surfaces using deep learning while preserving native geometry and topology of the cortical representation. It works by operating entirely in native 3D cortical space through diffeomorphic deformation. The framework consists of two main components: a Canonical Surface Generator (CSG), which builds a regularised mesh shared across subjects through simplification, subdivision, and deformation of a template mesh; and a Cortical Transformer, a dual-pathway architecture combining global and local attention mechanisms for feature extraction. The model is evaluated on phenotype prediction, fMRI activation regression, and ADOS score regression tasks, showing competitive performance compared to previous surface-based methods.

**Strengths:**

The paper aims to address an important problem which is the use of surface based models for cortical analysis to preserve the extraction of cortical signals and metrics while accounting from the geometry of the cortex.

The methodology introduced is relevant in the field: the need for cortical deep learning models that works on the native geometry rather that using template representations. It is an attempt to bypass standard spherical registration steps used in FreeSurfer (Fischl 2012) or HCP (Glasser et al 2013) pipelines for instance.

The method is experimenting on many datasets and against various models.

**Weaknesses:**

Despite an interesting methodology, the paper suffers from many shortcomings in its current state:

1. There is a limited contextualisation of the method within the neuroimaging field:
- In particular, there is no explanation of the reasoning behind the use of sphericalisation and spheres in neuroimaging pipelines (for visualisation, interpretability, registration, alignment), which goes beyond there usage for deep learning, and missing mention of established frameworks for comparing subjects (MSM, MSMAll, Robinson et al. 2018, Glasser et al. 2016). As stated it seems that the sphericalisation appeared in order to defined grid-like deep learning architectures (as stated in the intro).
- There is no proper related work sections introducing the previous surface deep learning methods and what advances they introduced compared to prior cortical analysis methods.
- No reference to prior self-supervised surface deep learning method using masked auto-encoding (MAE) methodologies (https://openreview.net/forum?id=9G7ZEYHLVJ).
- There no clear mention of prior work that tokenise biological surfaces (brains data e.g. Dahan et al 2022, Surface Vision Mamba He et al 2025, cardiac data e.g. LaB-GATr - Suk et al 2024)

2. The method is difficult to fully appreciate because the notion of distortion is neither clearly defined nor quantitatively evaluated. It remains unclear what type of distortion the authors are referring to - they probably refer to the sphericalisation process from native space to template space, thus should position against standard registration pipelines (FreeSurfer, HCP). However, by referring and comparing directly for instance to Dahan et al. 2022, it creates an ambiguity as that method operates already in a template space where the indexing between spherical and inflated surfaces is one-to-one, so the underlying cortical signal is preserved and no additional distortion is introduced. This poses a main issue in the positioning of the paper in its current writing.

3. There is no evaluation of the computational cost of the method (Canonical surface generator), which could be heavy for large datasets

4. There is an important lack of clarity regarding the evaluation procedure (benchmarked models) and the presentation of results.
-  Table 1 presents results with different representation of the data (Left, Right, Bilateral), without any mention if this choice comes from the benchmarked models or not. There is no explanation of what bilateral means? It would be more fair to run the benchmark on the same data configuration probably both R and L independently; or explain why some methods could not. Some methods use single hemispheres for predictions but are presented here with bilateral (e.g. Dahan et al 2022).
- More importantly, there is no explanation on how the benchmarking is done. Is the data processed similarly (via the Canonical surface generator) for all models or are the methods used with their own data representation? If the latter is correct the conclusions drawn in section 3.2 does not hold as there are two factor of variability (both methods and data representation).

5. No quantitative evaluation of the claimed “distortion reduction.” The only geometric validation compares to generic mesh algorithms, not to spherical or barycentric resampling.

6. Figures lack legends and are not self-explanatory; some methods are difficult to interpret visually.

7. There is no explanation or rational behind the phenotype prediction tasks selected in the HCP dataset, although the most popular phenotype prediction task remains scan age prediction in the literature (Fawaz et al 2021).

8. It is not clear what are the input cortical metrics used for the different tasks, only averaged fMRI is mentioned for functional activation map regression. Also why not using the same models for that task than for the phenotype prediction task?

9. Barely any discussion and limitations of the current approach.

**Questions:**

1. One of the most important questions concerns how the model handles non-diffeomorphic variations in cortical topology, for instance, folds that are present in some subjects but absent or split in others. Averaging across subjects to build a template mesh (Figure 1) likely removes these structures, raising the question of how subject-specific information is preserved. As shown in Glasser et al. 2016, it is not possible to register every individual cortex diffeomorphically, since cortical folding can vary in topologically inconsistent ways. Then, how does the proposed Canonical Surface Generator reconcile this fundamental anatomical variability with its diffeomorphic deformation framework?

2. How are patches defined in practice? it is not clear if they are manually defined, comes from contiguous regions or randomly defined?

3. what is the impact of the size of the patches?

4. How much inter-subject variability remains after the surface generation?

5. There appears to be a duplicated or incorrect citation for Dahan et al. 2022; please clarify whether this refers instead to this published work of Dahan et al 2022: https://proceedings.mlr.press/v172/dahan22a.html

---

### Meta-Review · Area_Chair_VzKn · 2026-01-06

**Summary:**

Reviewers agree the paper tackles an important problem in cortical surface deep learning: enabling Transformer-style representation learning directly on native cortical geometry, rather than relying on spherical projection. The proposed pipeline is viewed as conceptually relevant and potentially useful for building a shared topology/tokenization across subjects while preserving geometric detail. However, the main concerns are substantial and center on scientific positioning and evidential clarity: (i) the manuscript does not sufficiently contextualize "why sphericalization exists" in neuroimaging pipelines and how the proposed approach relates to established subject-comparison/registration frameworks; (ii) the claimed distortion reduction is not clearly defined nor quantitatively evaluated against the right reference points, and the group-average template/CSG may itself introduce alignment/geometry compromises; (iii) the benchmarking protocol is unclear and potentially confounded by inconsistent data configurations (left/right/bilateral) and uncertain preprocessing parity across methods; (iv) computational cost is not reported despite an apparently heavy multi-stage pipeline; (v) the related-work coverage is incomplete (including prior tokenization and self-supervised surface learning), weakening novelty/positioning; and (vi) important scientific questions remain about anatomical validity under inter-subject topological variability, patch construction, and bias/fairness factors known in cortical datasets. Given the lack of a substantive rebuttal, these issues remain unresolved and collectively support a reject recommendation despite the promising direction.

**Reviewer Concerns:**

Addressed by rebuttal:
* None in a meaningful way. The only author response is an acknowledgement/thank-you without technical clarifications or added evidence, so the review-critical points remain unanswered.

**Reviewer Scores:**

N/A, For this paper, the authors did not participate in the review process in a meaningful way.

---

### Decision · Program_Chairs · 2026-01-26

Reject